# Wearable and Stretchable SEBS/CB Polymer Conductive Strand as a Piezoresistive Strain Sensor

**DOI:** 10.3390/polym15071618

**Published:** 2023-03-24

**Authors:** Thaiskang Jamatia, Jiri Matyas, Robert Olejnik, Romana Danova, Jaroslav Maloch, David Skoda, Petr Slobodian, Ivo Kuritka

**Affiliations:** 1Centre of Polymer Systems, Tomas Bata University in Zlín, Tr. Tomase Bati 5678, 760 01 Zlin, Czech Republic; 2Department of Physics and Materials Engineering, Faculty of Technology, Tomas Bata University in Zlín, Vavrečkova 5669, 760 01 Zlin, Czech Republic; 3Department of Chemistry, Faculty of Technology, Tomas Bata University in Zlín, Vavrečkova 5669, 760 01 Zlin, Czech Republic

**Keywords:** highly stretchable, wearable electronics, SEBS/CB composite strand, novel preparation technique, piezoresistive sensor, respiration activity, strain posture detection

## Abstract

A wearable and stretchable strain sensor with a gauge factor above 23 was prepared using a simple and effective technique. Conducting nanocomposite strands were prepared from styrene-b-(ethylene-co-butylene)-b-styrene triblock copolymer (SEBS) and carbon black (CB) through a solvent-processing method that uses a syringe pump. This novel nanocomposite preparation technique is a straightforward and cost-effective process and is reported in the literature for the first time. The work included two stages: the flexible nanocomposite preparation stage and the piezoresistive sensor stage. Depending on its molecular structure, the thermoelastic polymer SEBS is highly resilient to stress and strain. The main aim of this work is to fabricate a highly flexible and piezoresistive nanocomposite fibre/strand. Among the prepared composites, a composite corresponding to a composition just above the percolation threshold was selected to prepare the strain sensor, which exhibited good flexibility and conductivity and a large piezoresistive effect that was linearly dependent on the applied strain. The prepared nanocomposite sensor was stitched onto a sports T-shirt. Commercially available knee and elbow sleeves were also purchased, and the nanocomposite SEBS/CB strands were sewn separately on the two sleeves. The results showed a high sensitivity of the sensing element in the case of breathing activity (normal breathing, a 35% change, and deep breathing at 135%, respectively). In the case of knee and elbow movements, simultaneous measurements were performed and found that the sensor was able to detect movement cycles during walking.

## 1. Introduction

The use of sensors has recently garnered significant attention, and its importance and need continue to grow. Sensors are transducers that transform energy from one domain into another. They convert various forms of stimuli, such as mechanical, optical or chemical signals, into electrical signals. Depending on their sensing mechanism, sensors can be classified as optic, inductive, piezoelectric, or piezoresistive. The piezoresistive sensors provide a high sensitivity with the simplest device design. Their working principle is based on the change in a material’s electrical properties when it is subjected to mechanical deformation such as strain or pressure [1,2]. More precisely, the change in resistance is measured when the material flexes through exposure to pressure or strain [3].

Conventional piezoresistive sensors tend to be rigid with limited mechanical flexibility. The need exists for more versatile and robust electronic gadgets in the form of healthcare monitoring devices and lighter wearable electronic systems. Therefore, the technology of the coming era will witness the incorporation of electronic systems with more robust technologies that integrate electronic devices into textiles, body joints, and skin. Fibre-like, flexible sensors are advantageous as they can be integrated into textiles as wearable sensors [4,5,6,7,8,9,10]. Wearable electronics demand that sensors be integrated into textiles by either weaving or sewing the sensor into the fabric. Since the sensor is part of the fabric, it can be bent, twisted, and stretched mechanically. Strain and pressure sensor functions are necessary to convert such mechanical stimuli into electrical signals. Moreover, electromechanical sensors with a high resistivity are desirable for recording real-time mechanical stimuli. As a result, a flexible, insulating matrix filled with conductive fillers makes for an efficient, stretchable piezoresistive sensor.

A matrix-filler system is used to achieve such a piezoresistive sensor. This nanocomposite system constitutes an organic (matrix) and an inorganic phase (filler) [11,12]. A polymer matrix was chosen for its stretchable and flexible nature. Styrene-b-(ethylene-co-butylene)-b-styrene triblock copolymer (SEBS) is one of the most commonly used thermoplastic elastomers (TPEs) due to its highly stretchable nature, processability, thermal stability, and lack of a need for vulcanisation [13,14,15,16,17]. Furthermore, several successful studies have been carried out on the incorporation of carbonaceous conductive fillers into SEBS matrices due to the growing interest in nanomaterials and nanotechnology. The nano-reinforcement of a polymer with carbon nanotubes (CNTs) and multiwalled CNTs (MWCNTs) has been extensively studied in the past few years [18,19,20,21,22]. Two-dimensional nanofillers were also employed, namely, graphene (GP) [23,24]. Moreover, combinations of CNTs and GP with zero-dimensional fullerene were investigated [25,26]. There have been extensive studies conducted on SEBS and filler nanocomposites with varying applications. Hofmann et al. blended SEBS with functionalised graphene (FG) and dispersed it in tetrahydrofuran (THF) using a melt-processing method. The SEBS/FG nanocomposites exhibited superior mechanical properties and an enhanced electrical conductivity at a lower percolation threshold. This nanocomposite has potential applications in the automotive and sealant industries [27]. Turgut et al. [28] utilised carbon nanofibres (CNFs) and SEBS to prepare nanocomposites. Similarly, CNF and an oil-swollen SEBS polymer were blended to prepare another set of nanocomposites. As expected, the increase in the filler content in both nanocomposites was accompanied by an improved tensile modulus and electrical conductivity. However, the former nanocomposites displayed a well-defined percolation threshold. The study also indicated that such composites have potential applications as strain sensors due to their high sensitivity.

Carbon black (CB) is a traditional and frequently used carbonaceous filler which does not lose its importance, is widely available in various forms, and has properties that are well known, investigated, and standardised. The addition of CB to nanocomposites with a SEBS matrix improves electrical properties and reinforces mechanical and viscoelastic behaviour [29]. In one study, Yang et al. prepared SEBS/CNT/CB nanocomposites [30]. In this work, a different volume % ratio of the fillers was investigated, and their sensing behaviours were analysed. The one-dimensional CB and the two-dimensional CNT complemented each other. CB particles filled the gaps or spacing between the conductive pathways from the CNTS. In another study, Kuester et al. [31] composed SEBS/EG (EG—expanded graphite) and SEBS/CB separately through melt-mixing, realising their application in electromagnetic interference shielding (EMI-SE). It was found that SEBS with 15 wt.% CB showed promising results for use in EMI-SE shielding applications, meaning that an elastic SEBS/CB composite with a reasonable electrical conductivity over a relatively large extent of deformation can be prepared to avoid unnecessary use of expensive nanofillers. This may be important for the safety and acceptance of these composites [32].

Moreover, wearable sensory devices are an innovative technology that has shown potential application in the field of electronic skins, such as as in human health monitoring, soft robotics, and various other human–machine interfaces [33,34,35,36,37,38,39]. These futuristic smart devices are designed to possess a high sensitivity, high flexibility, stretchability and wide detection range. Together with a thermoplastic polymer, they form an excellent combination for use as a piezoresistive sensor. In general, sensors of this type are utilised to detect body movements, and their largest advantage is the accuracy in measuring specific postures and movements. Based on the response from the individual sensors, it is possible to determine what kind of movement is executed and measure its intensity. There is an opportunity to use the produced SEBS/CB polymer sensor in the field of virtual reality. In virtual reality, the accuracy of movement detection is an important factor that affects the final quality of a game or other application. One of the complex detection tasks, especially during walking, is determining which foot is in front and which is behind [40,41,42].

Therefore, we propose the preparation of a SEBS/CB polymer conductive strand as a piezoresistive strain sensor. The advantage of our sensor lies in the direct application of the SEBS/CB polymer nanocomposite to clothing or parts of clothing. Furthermore, in using such an incorporated solution, motion detection is more accurate and can sense and detect body movements such as slow walking, brisk walking, and jogging without the need for the user to hold the monitoring device firmly in their hands or stand on it.

A specific application area for the prepared SEBS/CB polymer strand can be found in the rehabilitation techniques used in rehabilitation exercises. During rehabilitation exercises, it is very important to obtain feedback on the progress of the rehabilitated body parts [43]. In the form of a strand, a sensor is able to evaluate, for example, the bending of the limbs in the joint. Another advantage is that the weight of the SEBS/CB polymer strands used in the experimental measurements is low and therefore has a minimal effect on the load on muscles and joints. Respiration activities can also be detected during the rehabilitation process. Breathing can also refer to the oxygenation of the blood, which is one of the important indicators for medical doctors [44]. The breathing rate can also be easily measured. Identifying types of breathing can also refer to the activity performed by patients for deep breathing during heavy physical tests.

To summarize our aim and motivation, the work in this article is centred on the preparation of a flexible, conductive nanocomposite strand, SEBS/CB. An economical and eco-friendly solvent-processing method that uses a syringe pump extrusion without further heat treatment is the highlight of the nanocomposite preparation method. Additionally, the flexible, conductive nanocomposite fibre was used as a piezoresistive sensor and was integrated into a sports T-shirt and elbow and knee sleeves. This setup successfully monitored three different breathing phases and posture motions, demonstrating the sensor material’s extreme versatility and usefulness.

## 2. Results and Discussion

### 2.1. Morphology (SEM)

A cross-sectional scanning electron microscopy (SEM) image shows the SEBS matrix and the carbon black particles. In Figure 1, the dark grey colour represents the SEBS polymer matrix, and the white dot particles represent the carbon black nanofiller. The solution-processing method of nanocomposite preparation is evident from the SEM micro-graphs. The discontinuity of the SEBS matrix can be observed in the white, granular fillers sitting on the matrix. The SEM images of the sample also depict that the filler, carbon black, was dispersed well, with some minor agglomerates. The spherical carbon black particles are spread with a homogeneous filler distribution and are incorporated into the SEBS matrix, showing a good filler–matrix interaction. This compatibility is evident from the conductivity achieved on the nanocomposite sample.

### 2.2. Percolation as a Factor for Piezoresistivity Sensing Material Selection

The electrical behaviour of a conductive composite follows the percolation theory, which states that conductivity increases significantly—by many orders—when a critical volume fraction is reached: the percolation threshold. Above this value, the composite behaves as a conductor [21]. The concentration of nanofillers, their dispersions, and their processing technique are the deciding factors for the conductivity of a nanocomposite. In a polymer nanocomposite reinforced with conducting fillers, it is vital for the fillers to seep through or percolate into the polymer matrix. The network of fillers in the matrix leads to the electrical conductivity of the composite. This increases the electrical conductivity of the nanocomposite. Moreover, the connectivity of the filler particles resulting in the conductivity of the nanocomposite is either through physical contact with minimal resistance or incomplete contact with a tunnelling barrier. In general, a nanocomposite behaves as a conductor primarily due to the tunnelling of electrons through the barriers within the conductive filler network. If the conductive filler percentage is increased, the filler particles are greater in number and they are closer to each other. On the contrary, if the conductive filler percentage is decreased, the filler particles are less in number and the distance between them increases. This emphasizes the role of the mean gap size between the primary aggregates or clusters of CB particles embedded within the insulating polymer matrix. The tunnelling or hopping of charge carriers across the gaps between particles governs the conductivity of carbon-black-filled polymers above the percolation threshold [45]. The application of a strain to such a material may change the distance between the particles and influence the connectivity of the filler network, thus influencing the interparticle tunnelling. Changes in the tunnelling barriers influence the resistivity of the material, which is macroscopically manifested as a change in the measured resistance. The change in resistivity is responsible for the piezoresistive effect in nanocomposites. If the particle concentration is too high, the material has a saturated conductivity, and its resistivity may change slightly with the deformation. If the concentration of the filler is too low, the material behaves as an insulator. It should be noted that a decrease in conductivity is expected for elongation in the tensile mode. Therefore, the optimum material for making a piezoelectric strain sensor is a material with a filler concentration that just exceeds the percolation threshold. To assess the percolation behaviour of the composites, the conductivity of the SEBS/CB samples was measured as a function of the weight of the carbon black filler concentration. In accordance with the percolation theory, the composites displayed three stages of conductivity—insulation, percolation, and saturation. It is eminent from Figure 2 that the nanocomposite was insulating in the 0–10 wt.% region. From above 10 wt.% to 50 wt.% is the percolation stage when the nanocomposite begins to behave as a conductor. Finally, the 50–70 wt.% region is the saturation stage, without any significant increase in electrical conductivity. For the SEBS/CB compound, 15% is the threshold at which the SEBS/CB behaves as a conducting material. Therefore, this composition was chosen for the preparation of the strain sensor.

In Figure 2, which represents the percolation of the SEBS/CB vs. the conductivity samples, it can be seen that SEBS/CB 3% and the SEBS/CB 5% values are in the same exponential order. This is reflected in the graph. This can be attributed to the fact that the carbon black had just begun to percolate into the matrix. These two points display the starting or initialisation of conductivity in the matrix. The SEBS/CB 7% and SEBS/CB 10% values represent the increase in the conductivity of the nanocomposite with the increase in the amount of the carbon black conductive filler. As we observe the increase in the CB concentration, we can see a gradual increase in the conductivity.

The shape of the graph can be clearly explained by the s-like shape of a percolation curve graph from the percolation theory (as explained in Section 2.2 of the manuscript). The points from SEBS/CB 3% to SEBS/CB 5% are the insulation region in which the percolation threshold has not been reached. The points from SEBS/CB 5% to SEBS/CB 50% represent the conductivity region. This is the region in which percolation is reached, and the samples act as a conductor. The final region is the saturation phase, where the optimum electrical conductivity of the SEBS/CB composite is reached.

### 2.3. Mechanical Properties

A mechanical test in the form of the stress vs. strain relationship was conducted on the SEBS/CB 15% sample at two different elongation rates: 50 mm/min and 100 mm/min. As can be seen from Figure 3, at the 50 mm/min elongation rate, the elongation at break was approximately 262%. The elongation at break was approximately 300% for the 100 mm/min rate. The graph clearly demonstrates the highly stretchable property of the piezoresistive sensor at different elongation rates. This characteristic can find practical applications in wearable electrical and medical devices.

### 2.4. Electromechanical Properties

The electrical and mechanical properties of the SEBS/CB 15% conductive strand were tested. From Figure 4, it can be seen that the investigation was carried out as a function of resistance vs. the strain % and the relative response of the material Δ*R/R*_0_ vs. the strain %. The measurements were carried out in increments of elongation of 5 mm. The time difference between each measurement was 10 s. The figure below displays how stretchable the sample can be, demonstrating a strain range of up to 110%. The inset demonstrates the exponential character of the observed dependence. Therefore, a small, limited strain ranging from 0 up to a maximum of 16% strain was chosen for use in further experiments to avoid hysteresis and irreversible damage to the material.

In the follow-up test, the SEBS/CB 15% sample underwent a similar electro-mechanical measurement at the same elongation rates of 50 mm/min and 100 mm/min. It can be seen from Figure 5 that the elongation rate of 50 mm/min reached the upper measurement limit of the multimeter at elongation above 107%. For the 100 mm/min rate, this point was achieved above 234%. This test substantiates the highly flexible nature of the conductive SEBS/CB 15% nanocomposite strand, importantly at different elongation rates. This characteristic is essential for application in the detection of body movement, which is studied in this work. 

The images depicted in Figure 6 illustrate the SEBS/CB 15% sample in (a) unstretched, (b) stretched, and (c) bent mode. This flexible and stretchable nanocomposite strand can be used to detect various human motions.

In another demonstration in Figure 7, the SEBS/CB 15% piezoresistive sensor was subjected to a cyclic loading–unloading test (lasting up to almost 5000 cycles). It can be seen from Figure 7 that the sensor exhibited a high flexibility and durability. Moreover, insets (a) and (b) in the figure show that the conductive SEBS/CB 15% strand demonstrated a good endurance limit, long working life, and stability. Finally, in inset (c), it is evident that the mechanical stimuli corresponded to the electromechanical response of the sensor in the form of a sine wave (1 Hz frequency) with ±3% strain.

### 2.5. Gauge Factor and Piezoresistivity Analysis

One crucial parameter that is considered in evaluating the performance of a sensor is its sensitivity in the form of a gauge factor (*GF*). Simply put, this is the ratio of the change in resistance per unit to the change in length per unit. The *GF* is usually used to evaluate the sensitivity of a sensor with strain variation. Mathematically, *GF* is defined by the ratio of the relative difference of the measured relative resistance change (Δ*R/R*_0_) to the applied mechanical strain (*ε*):(1)$$GF=\frac{\left(\Delta R/R_{0}\right)}{\varepsilon }$$
where *R*_0_ represents the initial specimen resistance and Δ*R* represents the resistance change, *R* − *R*_0_.

Figure 8a depicts the gauge factor vs. strain relationship for the nanocomposite strand SEBS/CB 15%. The *GF* ranges from 28 to 23 upon elongation from 0 to 15%. Moreover, the relation shows an almost linear piezoresistive response.

A gauge factor of a piezoresistive strand strain sensor may be expressed as
(2)$$GF=1+2\nu +\frac{\left(\Delta \rho /\rho_{0}\right)}{\varepsilon }$$
where *ν* represents Poisson’s ratio, *ρ*_0_ represents the initial material resistivity, and Δ*ρ* represents the resistivity change, *ρ* − *ρ*_0_. The last term is the simplest form of a piezoresistivity coefficient and is usually considered constant [46].

Elastomers are generally considered to be practically incompressible, with Poisson’s ratio value nearly equal to 0.5 (for a cylindrical shape). Although Poisson’s ratio depends on the chemical structure of the elastomer and on the filler concentration, these dependencies are very weak over a broad range of the filler content. Hence, the value of 0.5 is a sufficiently good guess in our case, especially with respect to the experimental error, which is shown later. If there is no piezoresistivity effect, the value of the *GF* is just 2, i.e., 1 + 2*ν*. Indeed, the value of *GF* is 2 for many materials, typically metals. The exception for this is platinum, which demonstrates a *GF* ≈ 6 [47]. It is evident that *GF* values above 20 indicate a strong piezoelectric effect comparable with that of semiconductors [1].

In order to analyse the piezoelectric effect, Equation (2) was rearranged, multiplying both sides by the applied mechanical strain (*ε*):(3)$$GF\cdot \varepsilon =(1+2\nu )\cdot \varepsilon +(\Delta \rho /\rho_{0})$$

The product *GF·ε* was then plotted against *ε* in Figure 8b. In no way does the slope of the line have a value of 2. Instead, a value of 22.7 ± 0.3 is found. A small positive intercept is also observed. This implies a linear dependence of the term Δ*ρ*/*ρ*_0_ on *ε*, as in the following equation:(4)$$\Delta \rho /\rho_{0}=\frac{d\left(\Delta \rho /\rho_{0}\right)}{d\varepsilon }\cdot \varepsilon +const={const}_{1}\cdot \varepsilon +{const}_{2}$$

Yielding an expanded equation:(5)$$GF\cdot \varepsilon =\left(1+2\nu +{const}_{1}\right)\cdot \varepsilon +{const}_{2}$$

From the above, it is evident that the linear component of the piezoresistivity coefficient, *const*_1_, is approximately 20.7, indicating a linear dependency of the material’s resistivity change on the applied strain. In other words, Δ*ρ*/*ρ*_0_ increases 20.7 times when *ε* = 100%. This observation explains the exponential shape of the electromechanical behaviour curve in Figure 4. On the other hand, the small intercept of *const*_1_ with the value of 0.11 ± 0.03 does not change with the applied strain. Therefore, it most likely corresponds to the contribution of a small, inevitable systematic error to the measured values due to the resistance of the measuring wires, contact electrodes, and contact resistances as only a two-point method was used for the resistance measurement with the aid of the digital multimeter.

### 2.6. Thermal Properties (TGA-DSC)

The thermogravimetric (TGA) analysis was measured in air, in the temperature range of room temperature to 1000 °C at a 5 °C min^−1^ heating rate. The weight loss was analysed in weight percentage. The TGA curves of pure SEBS, carbon black, and SEBS/CB 15% are plotted in the upper graph window of Figure 9. It can be observed that the pure SEBS sample followed a degradation with poorly resolved steps. The TGA of SEBS/CB 15% shows that its thermal resistance was increased due to the carbon black filler. The onset degradation temperature of the nanocomposite was increased by approximately 50 °C when compared to pure SEBS. As can be seen, the 15% loading of carbon black (by mass) is evident from the last mass loss step of the SEBS/CB 15%, which corresponds to the loading percentage of the filler in the matrix. The DSC thermogram is shown in the lower graph window and matches well with the TG steps observed for the composite and its components. No phase change is indicated in the material at expectable application temperatures. The polymer matrix degradation began at above 200 °C and proceeded in several steps. The slight jump in the DSC thermogram at ca. 380 °C may be associated with the evolution of a bubble from the molten polymer. The maximum mass loss rate was achieved at ca 433 °C, whereas the last degradation step of the polymer matrix ended at approximately 560 °C. The remaining 15% of the CB oxidized completely below 740 °C.

### 2.7. Piezoresistive Wearable Sensor

In order to study the usability of the SEBS/CB nanocomposite strand, the SEBS/CB 15% sample was stitched on a sports T-shirt, and three different modes of breathing were monitored: normal breathing, deep breathing and inhale–hold–exhale. The test was conducted on an individual wearing the body-fit tee with the stretchable fibre stitched around the chest of the clothing. The three different modes of breathing were monitored as an electrical resistance response to the sensor. The real-time reading of the sensor was observed and was used for ten cycles. In another instance, the nanocomposite fibre was sewn onto the knee and elbow sleeves. These sleeves were worn by the volunteer on the right elbow and right knee. A brisk walk was conducted to test the piezoresistive response of the SEBS/CB 15% conductive fibre. The idea was to apply the sensing element to the T-shirt, knee sleeves, and elbow sleeves with minor modifications, as shown in Figure 10.

The sample calibration was performed based on the initial value of *R*_0_ for each experimental sample. For all measurements, the samples with an initial value equal to or close to the *R*_0_ value were used. The results are shown in Figure 10 and Figure 11 as a percent-age change in the relative resistance.
(6)$$\Delta R/R_{0}=\frac{\left(R-R_{0}\right)}{R_{0}}$$
where *R*_0_ is the electrical resistance of the measured sample before the first elongation, *R* is the resistance during elongation, and strain is the relative change in specimen length.

Subsequently, the cloth and the sleeves with the sensor incorporated were worn by a volunteer. The sensor was pre-strained by 10% of the initial length of the sensor. The pre-strain of the sensor ensured the acquisition of relevant data from the sensor. On the ends, there were outputs for contacts connected to the datalogger via cables. The maximum change in electrical resistance was 25% with normal breathing. The maximum change in electrical resistance was 150% with deep breathing (Figure 11). The response of the sensor to electrical resistance was very sensitive to deformation, reversible, and able to detect breathing in real time.

The stabilizing effect on the resistance extension loops and the residual normalized resistance change was constant from the beginning. The structure of the curves remained more or less the same regardless of the number of deformation cycles. In addition, the shape of the recorded peaks and curves was maintained irrespective of the number of deformation cycles. This mechanical stabilization is advantageous for using the sensor as an elongating sensing element. In the next section, the application of the sensor to the knee and elbow is described in Figure 12. The sensor on the sleeves was pre-strained by 10% of the initial length of the sensor, similar to the T-shirt. The cavity with the sensor was located on the axis of the measured joint. The volunteer wore the sleeves with sensors, and the movement of the knee and elbow on an elliptical trainer was simulated.

## 3. Materials and Methods

### 3.1. Materials

Polystyrene-block-poly(ethylene-ran-butylene)-block-polystyrene (SEBS) powder, average Mw = 118,000 was purchased from Sigma-Aldrich (Sigma-Aldrich St. Louis, MO, USA), and carbon black, Super P^®^ Conductive 99% (carbon black) was purchased from Alfa Aesar (Alfa Aesar, Heysham, Lancanshire, UK). The solvent toluene (p.a.) was procured from PENTA (PENTA s.r.o., Prague, Czech Republic). The extrusions were conducted with a New Era NE-1000 one-channel (New Era Pump Systems, Inc., Farmingdale, NY, USA), programmable syringe pump. The knee and elbow sleeves were purchased from Shock Doctor (size: XS) (Shock Doctor, Taiwan), and the sports T-shirt (size: S) (Decathalon, France), was purchased from Decathlon.

### 3.2. SEBS/CB Nanocomposite Preparation

To prepare the 15 wt.% nanocomposite (denoted as SEBS/CB 15%), 1 g of SEBS and 0.15 g of carbon black were added separately to a 25 mL beaker. Toluene was separately added to the beaker containing SEBS and CB. The beaker containing toluene and SEBS was stirred in a magnetic stirrer at 500 rpm for 30 min. In another 25 mL beaker on a magnetic stirrer, CB and toluene were mixed together for two hours. This was followed by the addition of the SEBS solution to the beaker containing CB (about 8 h) to properly mix the composite dispersion. This dispersion was then poured into a Petri dish (15 cm diameter), and the composite was allowed to dry overnight at room temperature. The dried SEBS/CB 15% composite was peeled off with tweezers, mixed with toluene, and loaded into a 6 mL syringe. A New Era NE-1000 one-channel, programmable syringe pump was utilised for the extrusion of the composite. The syringe opening served as the extrusion die. The extrusion velocity was 2.5 mL/h. An illustration of the solution-processing nanocomposite preparation is depicted in Figure 13. The extruded SEBS/CB fibre/strand was dried overnight at room temperature. The diameter of the extruded strand was 1.65 mm on average.

Similarly, the remaining composites were prepared with varying CB filler concentrations, summarized in Table 1.

### 3.3. Scanning Electron Microscopy (SEM)

The SEBS/CB 15% strand was dipped in liquid nitrogen and fractured to achieve a flat surface of the sample suitable for SEM analysis. To avoid charge build-up and artefacts on the sample, the fracture surfaces were coated with a conductive metal (Au/Pd) with a sputtering device. The sputtering process created a very thin conductive layer on the sample surface. This process rapidly increased the sample conductivity, which was crucial for obtaining SEM images. It was followed by placing the sample on a metal stub using carbon tape. The Nova NanoSEM 450 (FEI) microscope (Thermo Fisher Scientific, Hillsboro, Oregon, USA) was used with a Schottky field emission electron source and operated at 0.02–30 keV.

### 3.4. Thermogravimetric Analysis (TGA)

The thermogravimetric analysis (TGA) was performed on a Setaram LabSys Evo (SETARAM Instrumentation, Caluire, France) instrument to analyse the SEBS polymer and SEBS/CB nanocomposite degradation. The test was carried out to quantify and verify the concentration of the carbon black filler. The composite strand was sliced into a small piece (10 mg) and placed in an Al_2_O_3_ crucible. The samples were heated in an air atmosphere (60 mL min^−1^) up to 1000 °C at a heating rate of 5 °C min^−1^.

### 3.5. Percolation Measurement

The percolation curve of the SEBS/CB nanocomposite strands was measured on a KEITHLEY 6517B, TEKTRONIX, INC (Artisan Technology Group, Champaign, Illinois, USA). Each sample was connected using the two-point method. The length between the clamps was 100 mm. All samples had the same diameter of 1.65 mm. The technique was implemented to measure the conductivity of the composites with different concentrations of carbon black filler, Table 1.

### 3.6. Mechanical and Electromechanical Test

The Testometric M350-5CT (Testometric, Lancanshire, UK) mechanical testing machine was employed to study the mechanical test of the SEBS/CB nanocomposite strands. Pieces were sliced from a long-extruded SEBS/CB sample at 10 cm. The cut 10 cm sample was equipped with two metal electrode holders placed at 5 cm distance apart. The sample was fixed at two terminals of the Testometric M350-5CT instrument and connected to a digital multimeter UNI-T 71C (UNIT-T, Dongguan City, Guangdong Province, China). The analysis was carried out by stretching the strand in 5 mm increments, and the registered change in resistivity was noted. The test continued until the flexible SEBS/CB sample disintegrated during the testing process, Figure 4. The cyclic response of the SEBS/CB15% was conducted on an Instron 8871. 

### 3.7. Sensor Test

A sensor test was performed using the Keysight 34980A (Keysight, Santa Rosa, CA, USA) multifunctional switch/measure unit data-logger device. The sampling rate was adjusted to 50 ms. For monitoring different modes of breathing, one channel was used. Similarly, for the elbow and knee movement measurement, two channels with the same 50 ms sampling rate were used. The data-logging system was able to measure these multiple channels simultaneously. Sensors were connected by wires with alligator clamps to eliminate transition resistance. The position of the breathing sensor was carefully chosen in order to place the sample on the most sensitive part of the chest. The reading of the breathing motion process consisted of two aspects: the chest deformation during the breathing process and the durability of the T-shirt under normal wear and the comfort during its usage.

The process of choosing a sensor location can be divided into two stages. First, the dimension of the chest was measured using a conventional method to obtain the percentage change during the breathing process. The simulation of the chest deformation was performed by Avatar software. The software was able to simulate the deformation and show the deformation of the chest (while inhaling and exhaling) and identify the most effective location of the sensor with respect to its wearability. A garment-design software, CLO 3D, was employed for the simulation of bodily movements and tensions on the sports T-shirt. The virtual, 3D person in this software is called the “Avatar”, and the personal dimension features can be tailored according to, for example, height, chest circumference, waist circumference, arm length, leg length, etc. The Avatar model shows the distribution of deformation from the front and back parts of the chest, Figure 14.

## 4. Conclusions

This work demonstrates two things: firstly, the nanocomposite preparation was successful using a solvent-processing method of mixing the SEBS and CB, casting, drying, and finally extruding using a syringe pump. After surveying the literature across journals, we can confirm that this extrusion of nanocomposite samples using a syringe pump is the first of its kind. In the following observation, the as-prepared SEBS/CB nanocomposite strand was highly flexible and conductive due to the presence of the superconductive carbon black filler. After preparing a series of SEBS/CB nanocomposites (Table 1), a percolation threshold was achieved at SEBS/CB 15%. Moreover, this sample also exhibited good stretchability and durability, with a loading–unloading cyclic response of up to almost 5000 (Figure 7). The conducting fibre could be stretched up to 21 cm from its original length of 10 cm, which is an elongation of approximately 110%, Figure 4. This makes it feasible to utilise the strand as a flexible piezoresistive sensor that is also wearable. The proof-of-concept was observed when the SEBS/CB 15% strand was stitched on a sports T-shirt, elbow sleeve, and ankle sleeve and three modes of breathing and bodily motion (walking) were successfully monitored and measured. The good sensing performance, with a *GF* larger than 23, was attributed to a strong piezoresistive effect (Figure 8b) due to the presence of the conductive filler, carbon black, in a properly selected concentration above the percolation threshold, and the thermoplastic copolymer, SEBS, which provided the flexibility and stretchability properties. Moreover, the piezoresistive coefficient depended linearly on the applied strain. We presented a key wearable sensor with applications that can be extended to healthcare, robotics, and smart textiles, with some possible scope for improvement.

In summary, the following are the highlights of our work with the SEBS/CB nanocomposite:The solution-processing technique for preparing the SEBS/CB sensors is the first of its kind;The high flexibility of the sensor measured up to 300% (Figure 3);The highest sensitivity of the SEBS/CB was achieved at a CB concentration of 15% (by volume);A wearable piezoresistive sensor was demonstrated with the SEBS/CB 15% sensor.

## Figures and Tables

**Figure 1 polymers-15-01618-f001:**
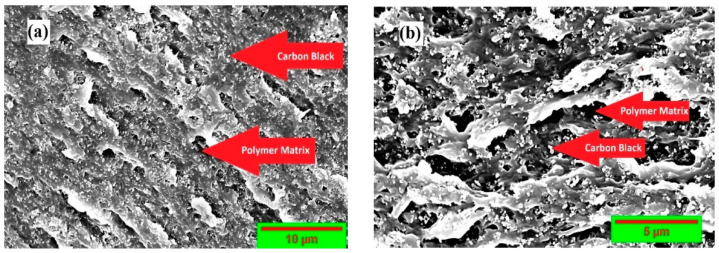
Scanning microscope micrograph of styrene-b-(ethylene-co-butylenes)-b-styrene triblock copolymer. (**a**) Surface morphology with CB nanocomposite; (**b**) cross-section of prepared nanocomposite.

**Figure 2 polymers-15-01618-f002:**
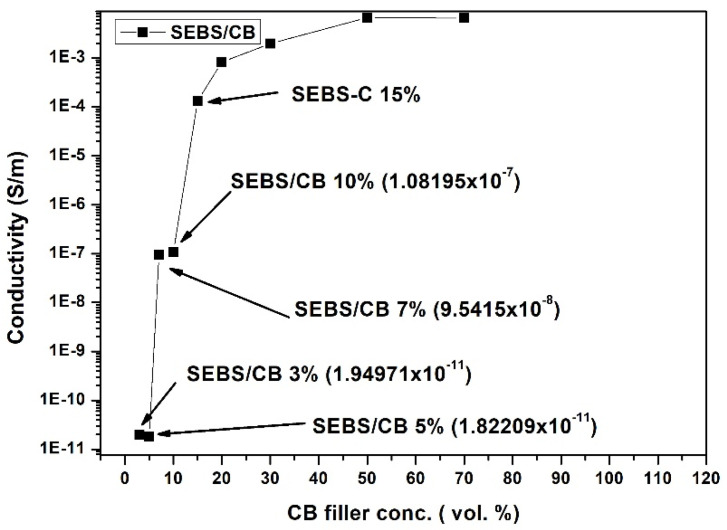
Percolation curve of SEBS/CB vs. conductivity. The arrow refers to chosen composite concentration for whole experimental study.

**Figure 3 polymers-15-01618-f003:**
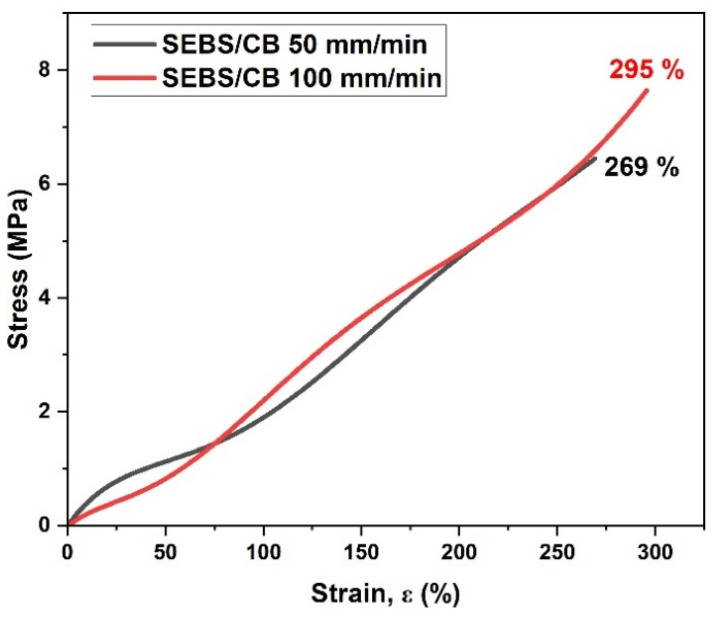
Stress–strain relation of the SEBS/CB 15% strand.

**Figure 4 polymers-15-01618-f004:**
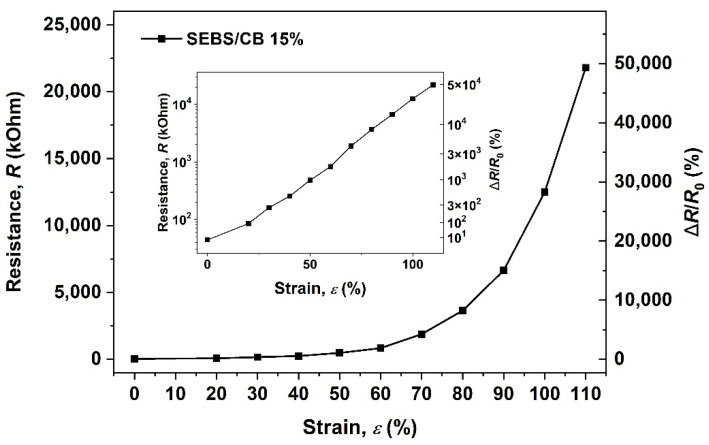
Electromechanical behaviour of styrene-b-(ethylene-co-butylenes)-b-styrene triblock copolymer, filled by 15 wt.% of carbon black (CB).

**Figure 5 polymers-15-01618-f005:**
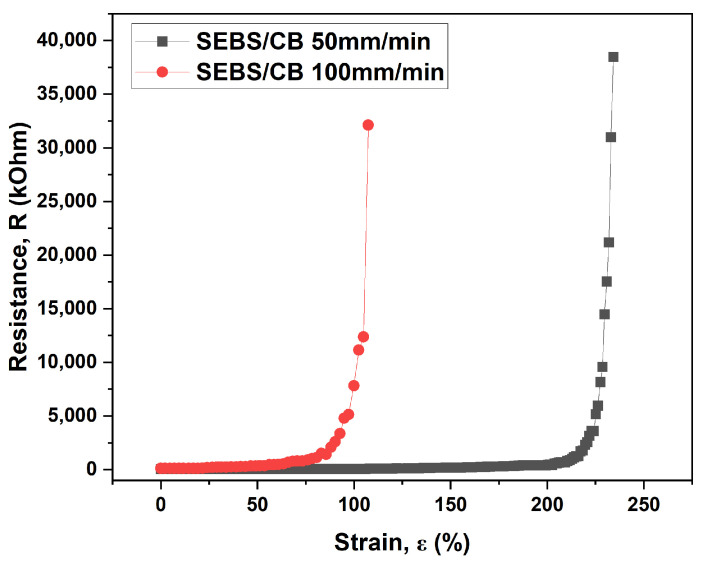
Electro-mechanical properties of the SEBS/CB 15% at different elongation rates.

**Figure 6 polymers-15-01618-f006:**
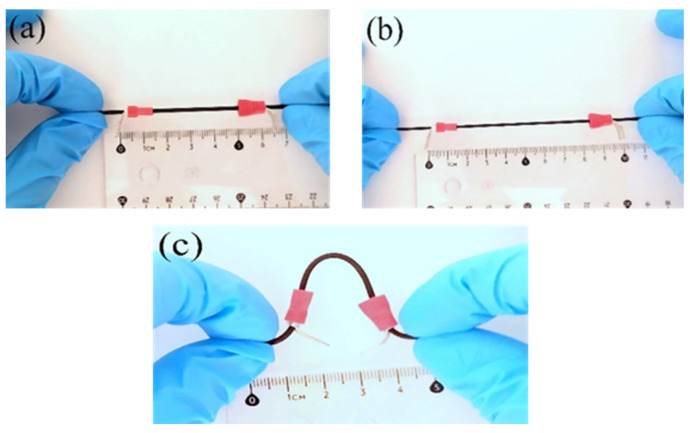
Pictorial representation of the SEBS/CB 15% sample in (**a**) unstretched, (**b**) stretched, and (**c**) bent form.

**Figure 7 polymers-15-01618-f007:**
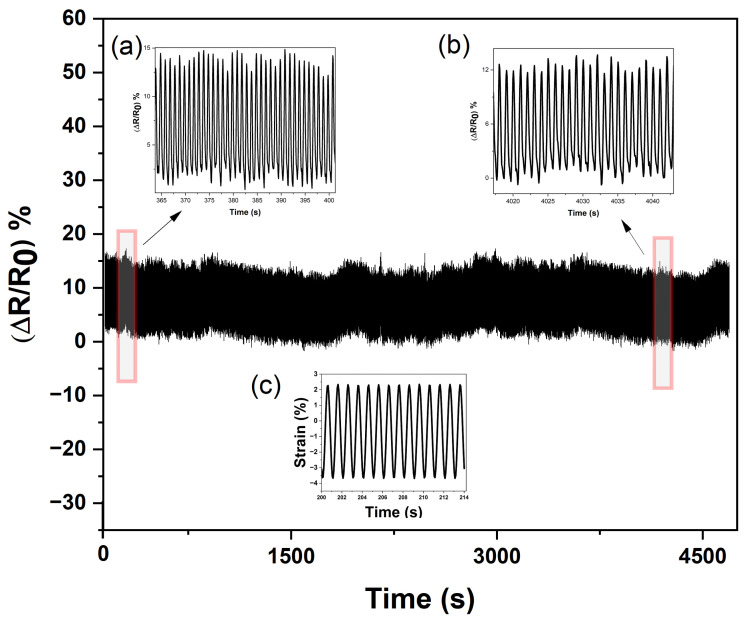
Durability and endurance test of the SEBS/CB 15% sensor. Insets (**a**,**b**) are the enlarged cyclic loading–unloading response, and (**c**) is the mechanical stimuli with respect to the electromechanical response.

**Figure 8 polymers-15-01618-f008:**
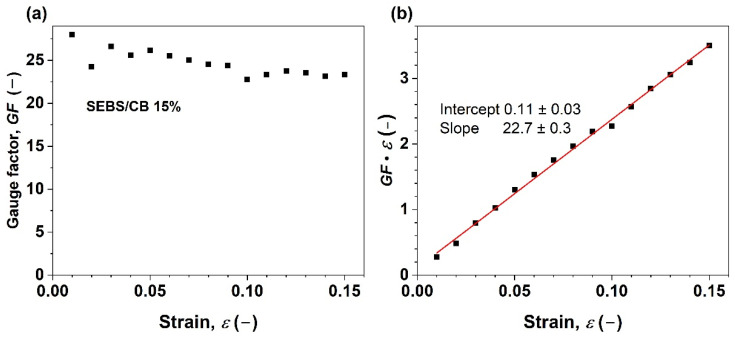
(**a**) Gauge factor vs. elongation of styrene-b-(ethylene-co-butylenes)-b-styrene triblock copolymer filled with 15 wt.% of carbon black (CB); (**b**) analysis of the resistivity change responsible for the piezoresistive effect. The red line represents a linear fit to the experimental data.

**Figure 9 polymers-15-01618-f009:**
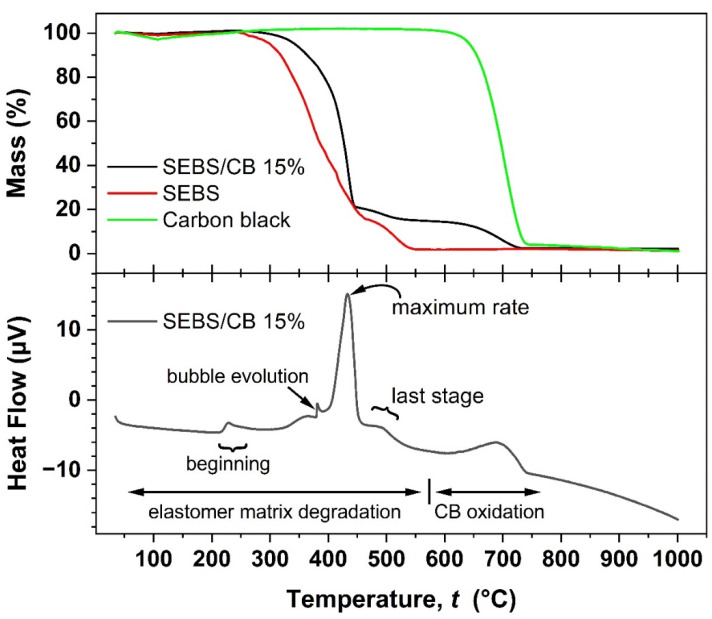
TGA analysis of SEBS, carbon black, and SEBS/CB 15% samples (upper graph window) and simultaneous DSC analysis of the SEBS/CB 15% sample (lower graph window).

**Figure 10 polymers-15-01618-f010:**
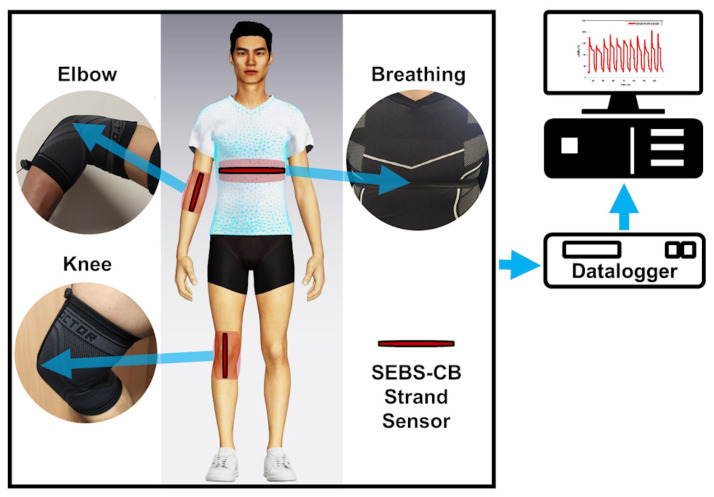
Schematic illustration of the position of the sensors on the chest, elbow, and knee.

**Figure 11 polymers-15-01618-f011:**
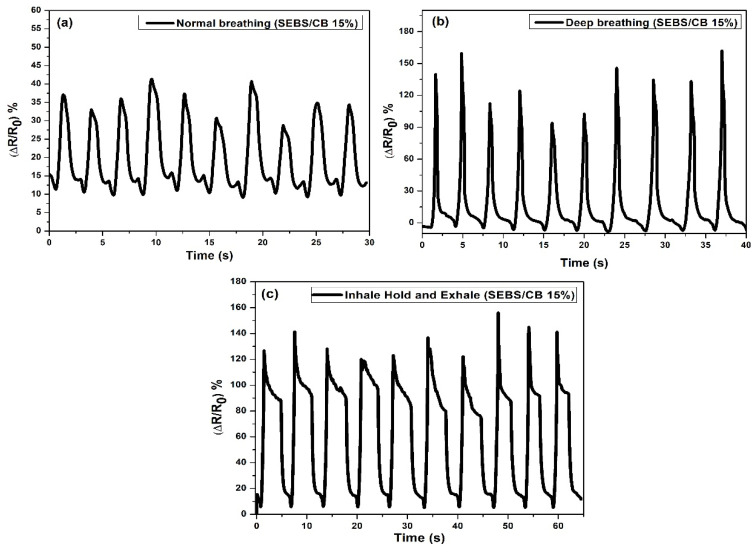
Different types of normal breathing represent by change of resistivity. (**a**) Normal breathing, (**b**) deep breathing, and (**c**) inhale–hold–exhale. Ten cycles were performed.

**Figure 12 polymers-15-01618-f012:**
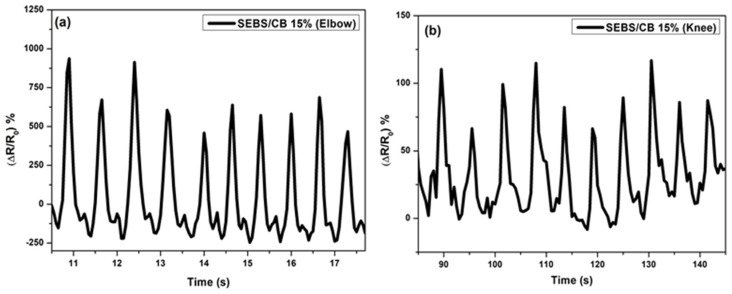
Change of resistivity vs. time for walking sensors: (**a**) elbow response and (**b**) knee response. The responses were measured simultaneously.

**Figure 13 polymers-15-01618-f013:**
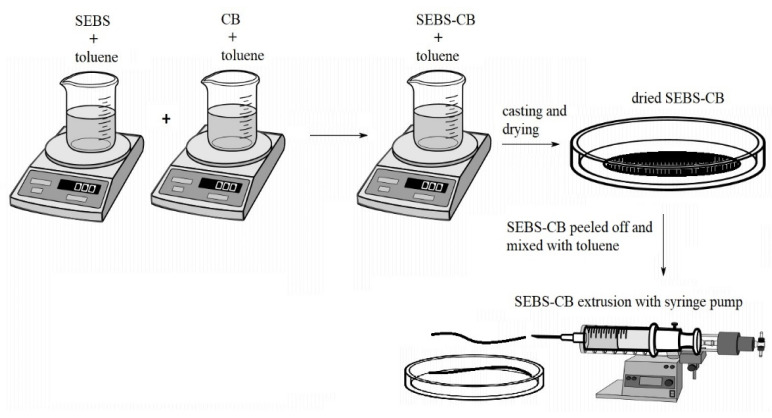
Illustration of a stretchable and flexible SEBS/CB strand preparation.

**Figure 14 polymers-15-01618-f014:**
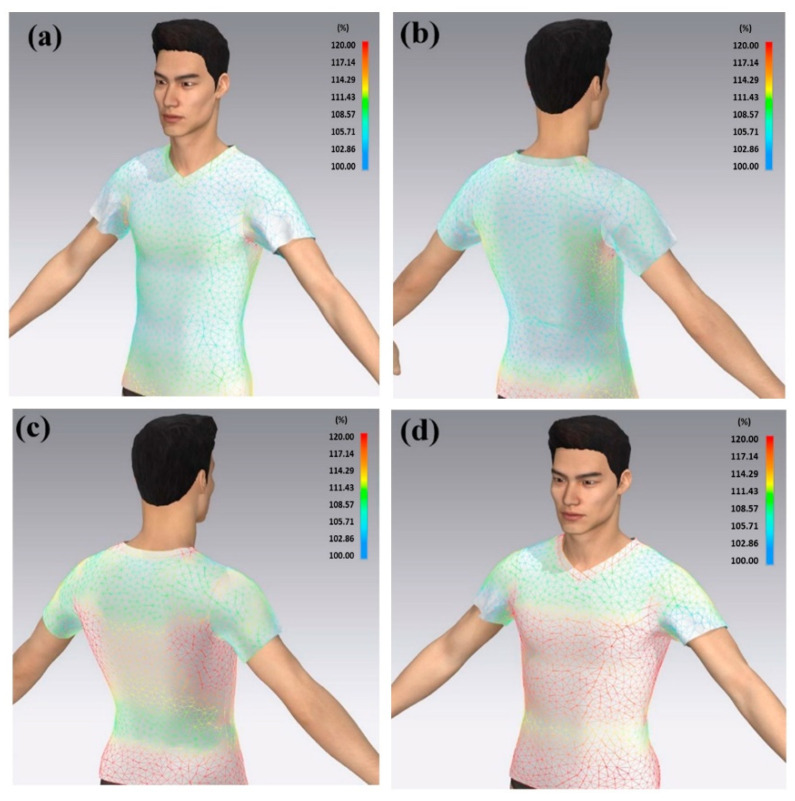
Tension simulation during breathing process. (**a**) Exhale—front view, (**b**) exhale—back view, (**c**) inhale—back view, and (**d**) inhale—front view. Colour scale shows the expansion of chest.

**Table 1 polymers-15-01618-t001:** SEBS/CB different filler conc.

Sample	SEBS (g)	CB (g)
SEBS/CB 3%	1	0.03
SEBS/CB 5%	1	0.05
SEBS/CB 7%	1	0.07
SEBS/CB 10%	1	0.10
SEBS/CB 15%	1	0.15
SEBS/CB 20%	1	0.20
SEBS/CB 30%	1	0.30
SEBS/CB 50%	1	0.50
SEBS/CB 70%	1	0.70

## Data Availability

Not applicable.

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
