# Peer review of "Wearable and Stretchable SEBS/CB Polymer Conductive Strand as a Piezoresistive Strain Sensor"

_polymers, 2023, doi:10.3390/polym15071618_

Round 1

Reviewer 1 Report

This work proposed a polymer conductive strand as a 108 piezoresistive strain sensor to clothing or parts of clothing. Furthermore, using such sensor in motion detection is more accurate and can sense and detect body movements like slow walking, brisk walking and jogging. The manuscript still has a few drawbacks as following:

(1) In addition to the influence of CB contents on the conductivity of composites, the stress-strain plot is also very important to decide the optimal CB content and to support the application as strain sensor. The authors should supplement this characterization.

(2) The test of strain sensing is too sample. The authors should supplement the tests of the relationship between strain and resistance under different stretching rates, response/recovery time, stability in cyclic loading/unloading, etc.

(3) Furthermore, several successful studies of SEBS/CB composites in strain sensors have been done, what is novelty and superiority of this work? Adding some explanations and comparisons. The potential applications of fibre-based strain sensors in human motion detection or health monitoring are widely studied, the authors should review some works in the part of Introduction.

Author Response

Dear reviewer

Reviewer 2 Report

1. In my opinion, in the Abstract section, the main challenge and concise perspective of this work should be included, and the redundant description of experiments should be further polished.

2. Annotation of the scale bar in Fig. 1(b) is missing.

3. The serial number of the figures should be the same. However, I found that just parts of the figures have brackets and others do not.

4. The scale bars in Fig. 11 are not clear to readers.

5. From Fig. 2, we can observe that the conductivities of polymers with 0-5 wt% and 6-10wt% SEBS/CB have little differences, respectively, why?

Reviewer 3 Report

It is an original paper dealing with “Wearable and Stretchable SEBS/CB Polymer Conductive Strand as a Piezoresistive Strain Sensor “supported by an experiment study. It is quite well organized and its language is quite satisfactory. However, there are some comments which should be described prior to acceptance for publication.

1.      The materials, preparation and measurement method for this sensor including sensor set up, equipment and the conducted tests, all must be clearly characterised and described  in different sections in advance before to start the result and discussion section.

2.      In Line 135, the author says ” SEM image shows the SEBS matrix and the carbon black particles” Please indicate the matrix and Carbon black particles by some arrows in the SEM pictures

3.      Dispersion method must be described already as you are describing it in the morphology section. If you have a picture of your dispersion method, please insert the picture in the Materials, Preparation and Measurement  method section

4.      In Fig.1(b) the magnification and scale is not clear

5.      In introduction the author explained the piezoresistve sensors and in the result and discussion, described the piezoelectric sensor. Actually, Piezoelectric is the property of a material to generate a voltage when mechanical force is applied to it. In contrast, the piezoresistive effect is the property of a material's resistivity to change when subjected to a mechanical force.

6.      What procedures did you follow to calibrate your electrical measurement set-up?

7.      In the material and method section, the author indicates toluene and acetone as Solvents (315 line). According to Fig.10, why the author was only beckon the toluene as a solvent?

8.      It would be helpful if you could list (bullet point) the most significant results

9.      It would be helpful if you address some relevant references to this research.

 In Line 41 please add two references below as you say “The working principle is based on the change of electrical properties of a material when subjected to mechanical deformation like strain or pressure” , and there is no references.

[a] Damage sensing of adhesively-bonded hybrid composite/steel joints using carbon nanotubes. Composites Science and Technology71(9), 1183-1189, 2011.

[b] MWCNT–epoxy nanocomposite sensors for structural health monitoring. Electronics7(8), 143, 2018.

In line 57, Please add the reference below after the sentence “This nanocomposite system constitutes an organic (matrix) and inorganic phase (filler).”

[c] Effects of nanoparticles on nanocomposites mode I and II fracture: A critical review. Progress in Adhesion and Adhesives3, 391-411, 2018.

In line 66, Please add a new reference ([c]) near the ref. 18

[d] Structural health monitoring of adhesive joints under pure mode I loading using the electrical impedance measurement. Engineering Fracture Mechanics245, 107585, 2021.

Round 2

Reviewer 1 Report

Dear authors, I think you should take more effort to improve your language, figures, and discussion. Specific suggestions are as follows: Firstly, I don't agree about your response about the repeatability and reversibility of this sensor. You'd better to supplement a long-term test about the resistance value under a fixed strain for at least 5000 cycles. This test can be found in many publications, e.g. https://doi.org/10.1016/j.cej.2021.133700. Secondly, in Fig.3 and Fig.5, with the same sample and same elongation rate, why the elongation at the break and strain are different? And the strain-stress plots in Fig.3 are strange, meaning the data is unstable. How to explain? Thirdly, because there are too many SEBS/CB or other SEBS/Conductive materials composites to be used as flexible or wearable strain sensors, what is your novelty and advantage? Try to make a comparison with other published works about mechanical properties, gauge factor (GF), stability, response time, working range, etc. I think you may draw a figure or a table to clarify it.

Round 3

Reviewer 1 Report

The authors have revised this manuscript carefully. I think this version can be accepted of the journal.